# Amplitude of circadian rhythms becomes weaken in the north, but there is no cline in the period of rhythm in a beetle

**Masato S. Abe[1], Kentarou Matsumura[2], Taishi Yoshii[3], Takahisa Miyatake[2]***

**1** Center for Advanced Intelligence Project, RIKEN, Tokyo, Japan, **2** Graduate School of Environmental and Life Science, Okayama University, Okayama, Japan, **3** Graduate School of Natural Science and Technology, Okayama University, Okayama, Japan

* miyatake@okayama-u.ac.jp

## Abstract

Many species show rhythmicity in activity, from the timing of flowering in plants to that of foraging behavior in animals. The free-running periods and amplitude (sometimes called strength or power) of circadian rhythms are often used as indicators of biological clocks. Many reports have shown that these traits are highly geographically variable, and interestingly, they often show latitudinal or longitudinal clines. In many cases, the higher the latitude is, the longer the free-running circadian period (i.e., period of rhythm) in insects and plants. However, reports of positive correlations between latitude or longitude and circadian rhythm traits, including free-running periods, the power of the rhythm and locomotor activity, are limited to certain taxonomic groups. Therefore, we collected a cosmopolitan stored-product pest species, the red flour beetle *Tribolium castaneum*, in various parts of Japan and examined its rhythm traits, including the power and period of the rhythm, which were calculated from locomotor activity. The analysis revealed that the power was significantly lower for beetles collected in northern areas than southern areas in Japan. However, it is worth noting that the period of circadian rhythm did not show any clines; specifically, it did not vary among the sampling sites, despite the very large sample size (n = 1585). We discuss why these cline trends were observed in *T. castaneum*.

## Introduction

Latitudinal clines are of evolutionary interest because they indicate the action of natural selection [1]. Many traits correlate with latitude, for example, body size [2, 3] and life history traits [4]. Circadian rhythm traits are also correlated with latitude [e.g., 5–8].

Circadian rhythms are particularly important for timing or regulating key biological events in insects [9]. The free-running period and power (i.e., sometimes called the amplitude or strength of the rhythm) are characteristics of rhythms [10, 11], and there is much evidence that the free-running periods of circadian rhythms exhibit latitudinal or longitudinal clines at the phenotype to molecular levels in many taxonomic groups [8, 12–21].

In insects other than *Drosophila*, many studies on the relationships between free-running periods and latitude have been conducted. In the linden bug, *Pyrrhocoris apterus*, higher-

**Funding:** This work was funded from Japan Society for the promotion of Science, Award number 16K14810 and 18H02510.

**Competing interests:** The authors have declared that no competing interests exist.

latitude populations are reported to have longer free-running periods [14]. Similarly, a positive relationship between latitude and period has been reported in a parasitic wasp, *Nasonia vitripennis* [15], and also in plant species [16].

In *Drosophila* species that evolved in tropical regions and then expanded their distribution to temperate regions, rhythm traits vary depending on latitude [17]. For example, pupal-adult eclosion rhythms in the far north were more arrhythmic than those in the south among *Drosophila littoralis* populations [18]. Circadian locomotor rhythms of *D. melanogaster* derived from Africa had a stronger power of rhythm than those of more northern *Drosophila* species in Europe [19]. The frequency of the *timeless* allele encoding a long TIMLESS (TIM) isoform, which has spread in Europe, studied from 40˚ to approximately 65˚in latitude was negatively correlated with latitude, i.e., higher frequency at lower latitude [5].

In addition, a pioneering study revealed a positive relationship between the strength of rhythm and latitude [7], while recent studies revealed a negative relationship between them [5, 6, 8]. Overall, these studies reveal that this topic remains controversial, and thus we will compare and then will discuss for the present results with other studies.

Given these discrepancies, we need more biological models. The red flour beetle *Tribolium castaneum* (Herbst) [Tenebrionidae] is a cosmopolitan stored-product pest [22], and hence, it can serve as an insect model insect species. Its biology and behavior are well studied [23, 24]. Furthermore, genome sequences are available for *Tribolium castaneum* [25].

This species is found throughout most of Japan, except for the northern island of Hokkaido, meaning that it can be collected in a wide range of latitudes [26]. It provides us with a novel, fascinating model with which to examine the relations between latitude/longitude and circadian rhythm traits. Hence, we studied the relationships between latitude or longitude and circadian rhythm parameters, including the period of circadian rhythms, power of the rhythm, and locomotor activity, in *T. castaneum*.

## Materials and methods

### Insects

The geographical populations of *Tribolium castaneum* used in the present study were collected from rice mills located at 38 different fields in Japan (Fig 1). S1 Table shows the latitude and longitude of the collection points, along with the number of samples. The northernmost and southernmost points where the insect could be collected were Aomori Prefecture (E127˚69' N40˚89'), and Kumamoto Prefecture-C (E130˚66' N32˚57'), respectively. The insects were collected during 2016 and 2017. We collected beetles from coin-operated rice pearling mills in Japan that were set up in villages or towns with rice fields. More than twenty beetles were collected from each mill.

The collected beetles were reared with a mixture of whole meal (Yoshikura Shokai, Tokyo) enriched with brewer's yeast (Asahi Beer, Tokyo) and maintained at 25˚C with a 16 h photoperiod (lights on at 07:00, lights off at 23:00). These laboratory conditions closely resemble natural conditions of this stored product pest. Each collected group was kept in a separate plastic Petri dish (90 mm in diameter, 15 mm in height). Before the experiments, each beetle population was reared for more than two generations in incubators (MIR-153, Sanyo, Osaka, Japan).

### Ethical note

The populations of *T. castaneum* used in this study were collected from each rice mill located at 38 different fields in Japan. The rice mills have signs saying "Feel free to take the rice bran." so we didn't need a permit to collect it from these mills. These laboratory conditions closely resemble natural conditions of this stored product pest. All individuals in the experiment were

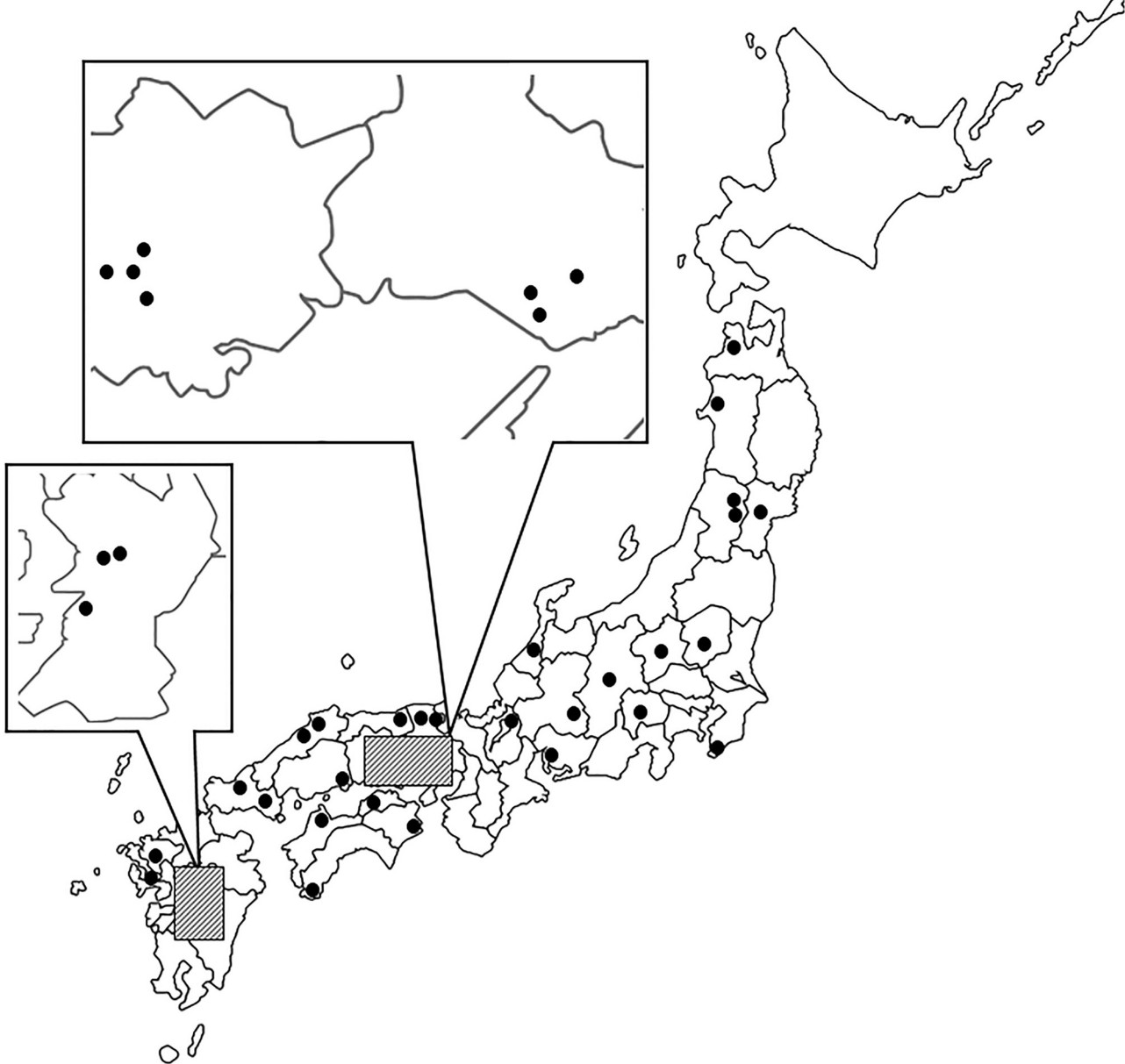

**Fig 1. Collection locations of wild *T. castaneum* populations in Japan.** This map was drawn from a free map of Japan on a net site (https://frame-illust.com/?cat=256), and thus it was not necessary to obtain permission.

handled with care and handling time was kept to an absolute minimum. The use of these beetles conforms to the Okayama University's Animal Ethics Policy.

## Locomotor activity

To assess circadian rhythmicity, we maintained beetles under 16L:8D conditions for more than 20 days in an incubator kept at 25°C before the measurement of locomotor activity, and we then measured the locomotor activity of *T. castaneum* for 10 days in darkness. A beetle from each population was placed in a clear plastic Petri dish (30 × 10 mm) in an incubator (MIR-153, Sanyo, Osaka, Japan) maintained at 25°C under complete darkness (DD). We

measured the circadian rhythms of 17 to 128 individual beetles for a single population. The sample size of each population measured for circadian rhythm is shown in S1 Table. The locomotor activity of each individual was monitored using an infrared actograph. An infrared light beam was passed through a clear Petri dish, and the beam was projected onto a photomicrosensor (E3S-AT11; Omron, Kyoto, Japan) that detected all interruptions of the light beam. Signals of interruption of the infrared light beam were recorded every 6 min [11]. At a time, we monitored 128 individuals, meaning we are measured 128 sensors at the same time.

## Statistical analysis

To determine the circadian rhythm, the locomotor activity data collected for 10 days in constant dark conditions were analyzed. The free-running period of circadian rhythms was established using a $\chi2$ periodogram test [27] for data on locomotor activity between 20 and 28 h [28]. Circadian rhythmicity was determined using $\chi2$ periodogram analysis, and "power" was used as an index of the strength of rhythms. The power of circadian rhythms was defined as the maximum difference between the $\chi2$ value and the significance threshold line at $P = 0.05$, that is, the size of the peak above the 5% threshold; see Fig 1 in [28]. Power is high when the rhythm is clear and strong, and a power of less than 0 indicates a statistically arrhythmic state. Moreover, total activity was calculated as the total number of interruptions of the infrared light over 10 days. To analyze the effects of latitude and sex on the period and power of the circadian rhythms and total activity, we used GLMs with Gaussian link functions. All statistical analyses were performed in R version 3.4.3 [29].

## Results

First, we analyzed the relationship between the geographical area and the power of the rhythm. To avoid multicollinearity between latitude and longitude (the correlation coefficient (r) between them was 0.83), we didn't include both simultaneously in the model as an explanatory variable but constructed a separate model with each value.

The GLM results revealed a significant relationship between latitude and power (Table 1, Fig 2A), while sex was not significantly associated with power (Table 1). The estimated coefficient for latitude in the model was negative, suggesting that the higher the latitude was, the weaker the rhythm was. For longitude, we did not find a significant relationship (Table 1, Fig 2B).

Second, we investigated the relationship between geographical area and the estimated period of circadian rhythms. The statistical results yielded no significant relationships between them (Table 1, Fig 3).

Finally, we investigated the relationship between geographical area and total activity (Fig 4). Because the total activity had some outliers, we used log-transformed values. The results of the regression analysis showed that total activity was associated positively with latitude and sex (Table 1, Fig 4).

## Discussion

In the present study, the power and period of circadian rhythms and total locomotor activity varied among geographical populations of *T. castaneum*. Circadian periods seemed to vary evenly between 20 h and 28 h (Fig 3). The present results showed that the power of circadian rhythms was significantly lower for beetles collected in northern areas than in southern areas (Fig 2). This result suggests that beetles collected from different parts of Japan have different characteristics. In this study, we reared individuals collected from the fields for a few generations in a chamber under the same environmental conditions in the laboratory before

**Table 1. Statistical test results.** Coefficients and standard errors obtained from GLM analysis are shown.

| | Response variable | | | | | |
|---|---|---|---|---|---|---|
| | Power of rhythm | | Period of rhythm | | log(Total activity) | |
| Latitude | -2.356* | - | -0.572 | - | 0.068** | - |
| | (-1.083) | - | (-0.435) | - | (-0.023) | - |
| Longitude | - | -1.126 | - | -0.288 | - | 0.023 |
| | - | (-0.629) | - | (-0.253) | - | (-0.014) |
| Sex (male) | -23.145 | -1.303 | 3.441 | 30.986 | 2.408* | 3.239 |
| | (-54.072) | (-118.738) | (-21.737) | (-47.701) | (-1.159) | (-2.549) |
| Latitude × sex (male) | 0.753 | - | -0.133 | - | -0.057 | - |
| | (-1.519) | - | (-0.611) | - | (-0.033) | - |
| Longitude × sex (male) | - | 0.037 | - | -0.238 | - | -0.021 |
| | - | (-0.876) | - | (-0.352) | - | (-0.019) |
| Constant | 54.626 | 123.33 | 257.799*** | 276.420*** | 4.523*** | 3.884* |
| | (-38.616) | (-85.312) | (-15.524) | (-34.273) | (-0.828) | (-1.832) |
| Adjusted R-squared | 0.003692 | 0.00331 | 0.001649 | 0.002568 | 0.02959 | 0.02588 |

* $p < 0.05$;

** $p < 0.01$;

*** $p < 0.001$

The hyphen represents the explanatory variable not included in the regression model.

measuring their traits. Therefore, it is unlikely that the present results were affected by any maternal effects.

Although clines in the power of the rhythm have been observed in only a few species, the trend of weaker circadian rhythms in northern populations has been observed in other insect species. Specifically, a clear rhythm was shown at lower latitudes whereas no rhythmic activity

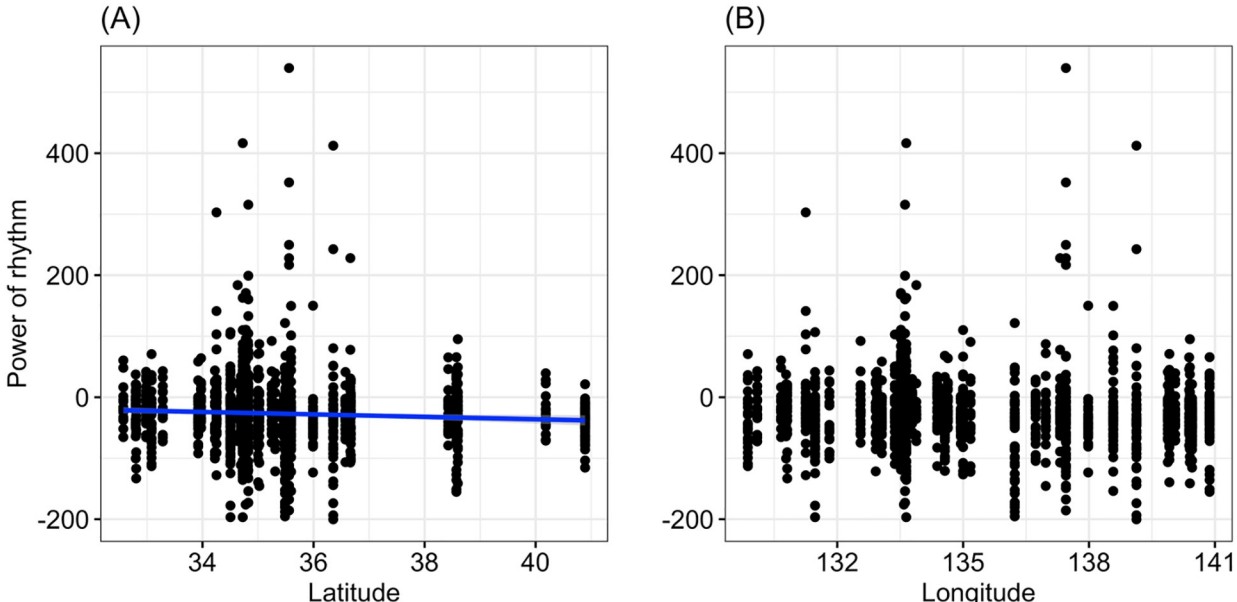

**Fig 2. Relationship between the power of the rhythm and latitude (A) or longitude (B).** The blue line represents the statistically significant regression line ($p < 0.05$).

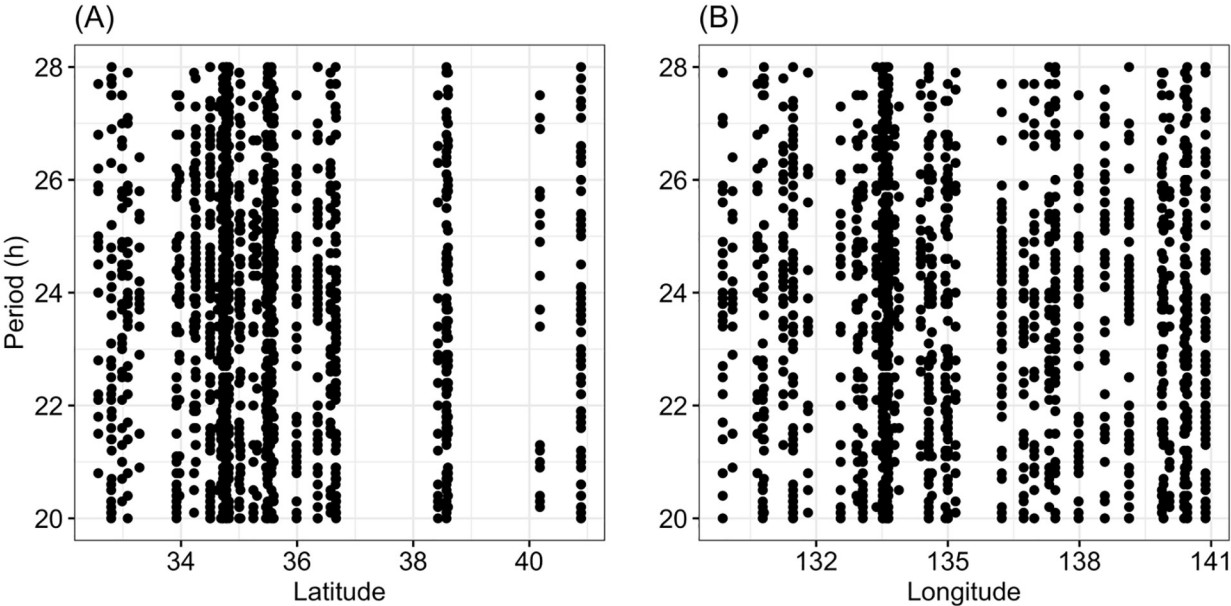

**Fig 3. Relationship between the period of the rhythm and latitude (A) or longitude (B).**

was shown at higher latitudes in *Hymenoptera* and *Drosophila* species [30, 31]. The present results are consistent with the results of these previous studies. Why is the rhythm weaker at higher latitudes? One answer may be that in more extreme environments, it may be easier to survive with less restriction of activity by the clock and more control by direct environmental responses, namely, masking of circadian activity [32, 33].

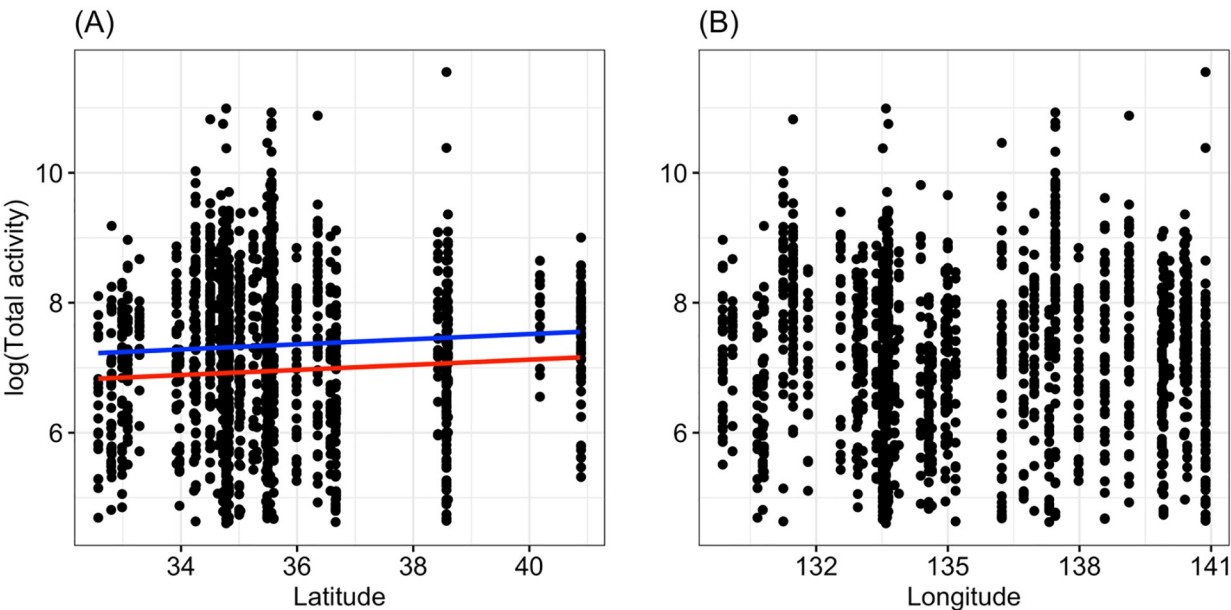

**Fig 4. Relationship between total activity and latitude (A) or longitude (B).** The blue and red lines represent the statistically significant regression lines of males and females, respectively ($p < 0.01$).

On the other hand, positive relationships between latitude and the length of circadian rhythms have often been reported; higher-latitude populations are reported to have longer free-running periods [14–16]. However, in the present study, no significant relationship was observed between the period of circadian rhythms and latitude or longitude, despite the very large sample sizes. The present study sampled beetles in a range of approximately 9 degrees latitude (from about 32˚N to 41˚N). Previous studies ([21] and references therein) have often measured rhythmic traits beyond the current range, but period of circadian rhythm has not yielded a distinct cline within the range of the present study; i.e., there was no effect of longitude or the longitudinal range (about 10 degrees). In other words, within this range of latitude, the range does not show a constant trend depending on the insect species examined [8]. Therefore, it is possible that a cline could be detected if beetle samples from more north were available. Because the northern limit of this species distribution is in Aomori Prefecture (about 41˚ degrees in latitude) where we examined this species, it may be possible to survey this point if we use a closely related species, such as *Tribolium confusum*, which may also inhabits Hokkaido. In comparing clock clines among species, caution may be required, because different *Drosophila* species have different clock networks [8]. Circadian rhythm is also affected by annual patterns of climatic factors including twilight duration, photoperiod, and temperature [6]. We didn't pay attention to the season when we were collecting beetles. However, our samples were maintained in a laboratory for two generations before use in the experiment, and thus we consider that such external weather effects could be ignored in this experiment.

The cline in the amplitude of the circadian rhythm in *T. castaneum* clearly suggests geographic variation. This result is very interesting considering the history of the controversy surrounding the dispersal distance of this insect, specifically, that studies do not agree on its dispersal characteristics. Some studies suggest that this beetle disperses very well [34]. Ridley et al. [34] inferred very high levels of active dispersal of *T. castaneum* through adult flight based on microsatellite genotypes. On the other hand, Drury et al. [35] and Arnold et al. [36] suggest limited dispersal for this beetle including short-range flying.

We considered two hypotheses regarding why no cline in the length of circadian rhythms was found in *T. castaneum*, as follows: bottlenecks and local adaptation. A small number of individuals or one fertilized female can enter and settle in individual rice mills scattered in the countryside [37]. A small number of *T. castaneum* will form each population within each mill. Predation pressures, including that from predator insects [38, 39], and the differences due to human cleaning will differ among mills. These pressures can cause differences in traits, especially in the activity traits, of *T. castaneum*. Such selection pressures (local adaptation) and founder effects (bottlenecks) would cause a large degree of variation among *T. castaneum* populations. *T. castaneum* cannot fly under stable conditions. Indeed, as described above, although these beetles have wings, dispersing by flying is rare at 25˚C [35], with walking being the most frequent mode of travel [36], and walking is the mechanism by which males locally search for females [39]. Additionally, Semeao et al. [38] showed that populations of *T. castaneum* collected from mills showed a spatial genetic structure, indicating the occurrence of a recent bottleneck in some mills. Therefore, each mill used in the present study may be an ideal system to clarify the mechanisms of bottlenecks and local adaptation of creatures in that mill.

On the other hand, adults of *T. castaneum* are known to fly well at temperatures above 28˚C [40]. There may be individuals flying beyond the area in the summer season in Japan. If gene flow is greater than expected, it might explain the lack of latitudinal and longitudinal clines shown in the present study. Ridley et al. [35] estimated the dispersal distance of *T. castaneum* using microsatellites and found that adults could fly at least 1 km per year in fields. They reported that *T. castaneum* aggregates predominantly around areas of grain storage but actively disperses by flight between these spatially separated resources [35].

Another study showed that genetic distance was not significantly correlated with geographic distance among *T. castaneum* populations in mills in the United States [38]. Semeao et al. [38] provided evidence that populations of *T. castaneum* collected from mills showed a spatial genetic structure, but the poor ability to assign individuals to source populations and the isolation at each mill suggested lower levels of gene flow than originally predicted. Konishi et al. [39] also suggested that different anti-predator strategies have evolved in each storehouse with and without predatory insects. These studies suggest that the dispute about gene flow or the possibility of evolution in individual storehouses in field mill systems will continue. The present result of a cline in circadian amplitude suggests that gene flow is not occurring on the spatial scale that we examined. However, estimation of the degree of gene flow between rice mills and phylogenetic relationships between populations within the species in Japan is required.

Notably, the results of the present study showed *P* values smaller than 0.05, even for small effect sizes, due to the very large sample size ($n = 1585$, total sample size). This suggests that the power of circadian rhythms is smaller at higher latitudes, but the difference is not large. Moreover, it is worth noting that the period of circadian rhythms does not change, even with the very large sample size, in this beetle species. It should also be noted that the negative relationship of power of rhythm to latitude, and the positive relationship of total activity to latitude in the present study may have detected significant effects despite the weak relationship due to the very large sample size (see Adjusted R-squared in Table 1).

Further studies on circadian rhythms at population and molecular levels (e.g., clines in alleles of specific circadian genes) are needed in the future [see 5, 6, 8, 17, 21]. Furthermore, given the seasonal differences in light exposure at different latitudes, studying the phase shift of the locomotor rhythm to brief light pulses might also provide a relevant phenotype that can discriminate between northern and southern population, as was the case for *Drosophila timeless* variants [21].

The frequency of the *timeless* allele encoding a long TIM isoform of *D. melanogaster* first studied from 40˚N to 65˚N was initially negatively correlated with latitude, but this trend was reversed in populations collected in extreme southern Europe (32–35˚N) [21]. It turned out that the latitudinal cline was actually due to natural selection spreading the recently derived *timeless* allele in all directions from a point of origin in southern Italy, thereby generating a distance rather than a latitudinal cline. This 'distance cline' was confirmed by studying the *timeless* allele in the Iberian Peninsula [41] where the cline was reversed. These studies reveal that the history of genes that encode clinal characteristics also needs to be considered when attempting to interpret spatial distributions. Such historical perspectives may account for some of the contradictory results that have been generated in the geographical analyses of biological rhythm phenotypes.

## Supporting information

**S1 Table. Collection site of *T. castaneum* populations with latitude and longitude.**
(XLSX)

**S1 Data.**
(XLSX)

## Acknowledgments

We appreciate Yusuke Tsushima and Kouhei Nakao for insect sampling.

## Author Contributions

**Conceptualization:** Takahisa Miyatake.

**Data curation:** Masato S. Abe, Kentarou Matsumura, Takahisa Miyatake.

**Formal analysis:** Masato S. Abe.

**Funding acquisition:** Takahisa Miyatake.

**Investigation:** Masato S. Abe, Kentarou Matsumura.

**Methodology:** Masato S. Abe, Taishi Yoshii, Takahisa Miyatake.

**Project administration:** Takahisa Miyatake.

**Software:** Masato S. Abe, Taishi Yoshii.

**Supervision:** Takahisa Miyatake.

**Validation:** Masato S. Abe.

**Visualization:** Masato S. Abe, Kentarou Matsumura.

**Writing – original draft:** Masato S. Abe, Takahisa Miyatake.

**Writing – review & editing:** Masato S. Abe, Kentarou Matsumura, Taishi Yoshii, Takahisa Miyatake.

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
