## [Decision Letter · Decision Letter 0]

9 Oct 2020

PONE-D-20-27518

Amplitude of circadian rhythms becomes weaker in the north, but there is no cline in the period of rhythm in a beetle

PLOS ONE

Dear Dr. Miyatake,

Thank you for submitting your manuscript to PLOS ONE. After careful consideration, we feel that it has merit but does not fully meet PLOS ONE’s publication criteria as it currently stands. Therefore, we invite you to submit a revised version of the manuscript that addresses the points raised during the review process.

In the discussion please elaborate on: What is the seasonal variation in photoperiod at the northern vs. southern end of the range? What are the typical temperature ranges? How does this compare with other studies in which circadian clines were or weren't found? 

Fix issues with the Methods as indicated by reviewer 2.

Discuss the Helfrich-Foerster and Costa studies on weaker fly circadian rhythms in northern latitudes mentioned by reviewer 1. 

Consider adding phase shift response data. 

Discuss (small) effect sizes.

We look forward to receiving your revised manuscript.

Kind regards,

Henrik Oster, Ph.D.

Academic Editor

PLOS ONE

Journal Requirements:

2. In your Methods section, please provide additional information regarding the permits you obtained for the work. Please ensure you have included the full name of the authority that approved the collection sites access and, if no permits were required, a brief statement explaining why.

3. Please include your tables as part of your main manuscript and remove the individual files. ** Please note that supplementary tables (should remain/ be uploaded) as separate "supporting information" files **.

"This work was supported by the Japan Society for the Promotion of Science Grant-in-Aid for Scientific Research Grants 16K14810 and 18H02510 to TM.".

i) Please provide an amended statement that declares *all* the funding or sources of support (whether external or internal to your organization) received during this study, as detailed online in our guide for authors at http://journals.plos.org/plosone/s/submit-now.  Please also include the statement “There was no additional external funding received for this study.” in your updated Funding Statement.

ii) Please include your amended Funding Statement within your cover letter. We will change the online submission form on your behalf.

5.  We note that [Figure 1] in your submission contain [map/satellite] images which may be copyrighted. All PLOS content is published under the Creative Commons Attribution License (CC BY 4.0), which means that the manuscript, images, and Supporting Information files will be freely available online, and any third party is permitted to access, download, copy, distribute, and use these materials in any way, even commercially, with proper attribution. For these reasons, we cannot publish previously copyrighted maps or satellite images created using proprietary data, such as Google software (Google Maps, Street View, and Earth). For more information, see our copyright guidelines: http://journals.plos.org/plosone/s/licenses-and-copyright.

1.     You may seek permission from the original copyright holder of Figure(s) [1] to publish the content specifically under the CC BY 4.0 license.  

Additional Editor Comments (if provided):

n/a

Reviewers' comments:

Reviewer's Responses to Questions

**Comments to the Author**

1. Is the manuscript technically sound, and do the data support the conclusions?

Reviewer #1: Yes

Reviewer #2: Yes

2. Has the statistical analysis been performed appropriately and rigorously? 

Reviewer #1: Yes

Reviewer #2: I Don't Know

3. Have the authors made all data underlying the findings in their manuscript fully available?

Reviewer #1: Yes

Reviewer #2: Yes

4. Is the manuscript presented in an intelligible fashion and written in standard English?

Reviewer #1: Yes

Reviewer #2: No

5. Review Comments to the Author

Reviewer #1: This is an interesting ms concerning the geographical variability of the circadian rhythm phenotype in T castaneum. The literature is mixed on this subject with classical Pittendrigh experiments in Drosophila species in Japan suggesting that the amplitude of the rhythm is stronger at higher latitudes because it has to counterbalance the effects of extreme photoperiods which would tend to make animals arrhythmic. There is not much molecular evidence to back this up, indeed recent studies from Helfrich-Foerster suggest the opposite, in that a weaker circadian rhythm in northern latitudes is more flexible for adaptation of circadian behavior to extreme photoperiodic conditions. Costa's group suggests that the circadian light response in D. melanogaster should be reduced at higher latitudes - this pair of papers (Science, 2007???) is not cited. Consequently it would have been nice if the study had also included a phase shift response to light as that would have been perhaps more relevant at these latitudes. Nevertheless, this study comes out in favour of the latter hypothesis. The results reflect a lot of work and the ms is clear to read and understand. My one quibble is that the effects are tiny and require a huge N to be observed. Also the variation in latitude is only 8oC which doesn't help to dissect out an effect. Furthermore the power statistic used is not very sophisticated and there are better methods available, but I suspect the result would be the same. The Discussion is a bit too long considering the brevity and conciseness of the results, so perhaps a little less speculation would be in order. Otherwise, the study is OK.

Reviewer #2: The authors conducted a study on the effects of geographic (primarily latitudinal) variation on circadian rhythmicity in the red flour beetle. Strengths of the study included the large sample size and rearing the populations for 2 generations in the lab to separate out maternal/environmental effects. While the English was professionally edited, I often found the ideas to be communicated in an imprecise or unclear manner.

1. It was completely unclear why the authors repeatedly referred to longitude early in the manuscript (e.g., L29, L40...a key word!, L47, L53, L78). The study is primarily designed to detect latitudinal clines given the much larger latitudinal range sampled. The authors state that longitudinal clines are also known (L47) but don't give a citation. Based on their titles, all the references cited on L54 (refs 8-12) refer to latitudinal clines. I can understand why they would want to test for this possibility within their data, but I don't know that there is any a priori reason to expect a longitudinal cline. This was also very confusing to me because they referred to altitudinal clines (L26 of abstract) making me think that maybe these would be investigated, but think they were ever discussed in the manuscript. A third point of confusion is that the authors said they were analyzing effects of temperature (L130), but I don't think they ever did.

2. The discussion on L61-69 is very unclear. The authors seem to be mixing points related to "where flies live" and latitude. By "where flies live" do they mean geographic features that might be distinct from latitude? If so what are they? L67 "pioneering"? Some of the studies cited here are quite old ("pioneering"). Is there a consensus emerging in the newer literature? The discussion overall was hard to follow.

3. The predominant finding of the paper was that there is an effect of latitude on amplitude/power of the circadian rhythm. This finding would be more useful if it were put into some kind of context. What is the seasonal variation in photoperiod at the northern vs. southern end of the range? What are the typical temperature ranges? How does this compare with other studies in which circadian clines were or weren't found?

4. While the sample size was apparently quite large, it was described in a very confusing way. For example, L109-110 makes it sound like only one beetle was measured per population. Maybe one was measured per population in each trial and several were monitored at once? L117/TableS1 is confusing because it sounds like it would be the number of beetles measured per population, but I think this is the starting population that was collected and used for breeding (??).

5. L113 the mention of "a clear Petri dish" again confused me...these are the same petri dishes referred to in L110, right? How many petri dishes were monitored at a time? Was there one animal in each dish? I was really confused.

6. L151-152. This description was inadequate. In what direction was the relationship, and how did it compare with the latitudinal effect on amplitude/power? I think this is important because total activity may affect the power to detect a change in amplitude or period (if animals aren't moving much it's harder to detect a change). Are these trends strong enough that such an indirect effect might be important?

7. L158 this statement is really vague. Are the authors talking about clines or just inter-individual variation? What are they trying to say?

8. L166-167 The transition to this sentence is really rough. "On the other hand" would usually indicate a contrast with the previous statement, which isn't the case here.

9. L176-177 This sentence seemed like an odd way to end the paragraph. And it is a sort of contrast to the opening sentence (L168-9). Why not combine them into one strong sentence at the beginning. Apparently, a limited number of studies have identified this trend. It's fine to just say that (once) and would be even better if the authors could put it in context. Are there just a few because it's only occasionally studied? Because most studies don't have sufficient power, or because it seems to be rare?

10. L188-194 This section again convoluted. L188-190 is a wordy sentence that should be streamlined/clarified. The examples should be clearly organized to indicate which support wide dispersal and which don't. My sense is that both Drury (33) and Arnold (34) suggest limited dispersal. This is obscured by the use of "on the other hand" which implies a contrast. On top of this the text about Semeao's study is repeated twice with very similar text (L195-197 and L233-237).

11. L252 I would argue that not only do they need molecular studies of the circadian rhythmicity (e.g., clines in alleles of specific circadian genes), but also to couple genetic studies (i.e. population assignment) more directly with phenotypic studies.

Minor

L51 "indicators of circadian rhythms" is vague. These are characteristics of rhythms, and variations in these characteristics could indicate adaptation etc.

L55 "Additionally" here doesn't make sense. The topic sentence makes it sound like they are switching to talking about "insects other than Drosophila" but the preceding paragraph is also mostly about insects and there is no mention of Drosophila.

L58-59 a positive relationship between "latitude and circadian rhythm" doesn't make sense. I think the authors might mean a positive relationship between latitude and period.

L75 The syntax is hard to follow. I suggest "This species is found throughout most of Japan, except for the northern island of Hokkaido, meaning..."

L89-91 Please write the coordinates in a more standard way.

L101-103 These two sentences almost completely repeat each other.

L106 "To assess circadian rhythmicity" or "To assess characteristics of circadian rhythms" or "To assess circadian phenotypes"

L123 periodogram is misspelled.

L137 power of the rhythm

L138-139 this description was confusing. It made it sound like they were only considering one (presumably latitude), but in fact they considered each of them in separate analyses.

L218 "if without human cleaning" doesn't make any sense. I kind of understand that humans cleaning out the pests would disrupt the colonization/selection/evolution, but that's not explained.

L236 I might misunderstand, but wouldn't lack of isolation suggest higher gene flow?

L242 suggest "is not occurring on the spatial scale that we examined"

L246 significantly smaller than what? I think the authors just mean significant p values.

L253 I don't know what the authors mean by different phenotypic phenomena than other insect species.

Table 1 was misaligned in my version.

6. PLOS authors have the option to publish the peer review history of their article (what does this mean?). If published, this will include your full peer review and any attached files.

Reviewer #1: No

Reviewer #2: No

---

## [Author Response · Author response to Decision Letter 0]

20 Nov 2020

11th November 2020

Dear Dr. Henrik Oster (Editor),

Thank you for allowing us to revise and so much for your many useful suggestions for our paper “PONE-D-20-27518, Amplitude of circadian rhythms becomes weaker in the north, but there is no cline in the period of rhythm in a beetle”, and for the two expert reviews. We have revised the manuscript as per the referee requests and hope the revised version is now suitable for publication in '' PLOS ONE”. We have addressed all of the reviewer’s comments – see our responses below. 

The author responses to the reviewer’s comments are as follows. Each reviewer’s comment is highlighted in bold, and our responses follow immediately below.

Many thanks,

Takahisa Miyatake

Editors comments:

In the discussion please elaborate on: What is the seasonal variation in photoperiod at the northern vs. southern end of the range? What are the typical temperature ranges? How does this compare with other studies in which circadian clines were or weren't found? 

We have added to our consideration of your point and discussed these as you requested (Lines 190-211).

Fix issues with the Methods as indicated by reviewer 2.

Author responses: We have fixed the issue raised from reviewer 2, described below.

Discuss the Helfrich-Foerster and Costa studies on weaker fly circadian rhythms in northern latitudes mentioned by reviewer 1. 

Author responses: Thank you for your teaching the two important papers which show opposite results from our study. We have cited these, and discussed with these studies (Lines 66-75 and190-198).　

Consider adding phase shift response data. 

Author responses: Thank you for your suggestion. We would have liked to do this if we could, but many of the samples we had already collected had been discarded and we were unable to conduct this experiment.　Also, we are currently using the measurement equipment for other projects and are sorry, but we are not in a position to conduct the experiments that the editor pointed out. Instead, we added the phrase “Furthermore, future experiments with phase-shifted experiments will provide a more detailed understanding of the relationship between circadian rhythms and latitude” at the last of Discussion (Lines 283-285).

Discuss (small) effect sizes.

Author responses: Added the discussion for small effect size in lines 269-278, and we have also added “Adjusted R-squared”, an index to show the effect size in the revised manuscript. 

Since there was a weak negative correlation between activity and power of rhythm (Spearman’s correlation coefficient = -0.22), lower activities do not necessarily cause lower rhythm.

Moreover, we conducted path analysis to detect the direct and indirect effects. We found the significant direct effect of latitude on power of rhythm and the similar estimated regression coefficient (-1.84 +- 0.76(SE)) to the one reported in the main text (-2.35 +- 1.08(SE)) although there was the significant indirect effect (latitude -> activity -> power of rhythm).

Therefore, we concluded that higher latitude caused lower rhythm although the effect size was small. Likewise, the result of the period was qualitatively same with the one in the main text.

'Response to Reviewers'

Dear reviewers; Thank you for allowing us to revise and so much for your many useful suggestions for our paper “PONE-D-20-27518, Amplitude of circadian rhythms becomes weaker in the north, but there is no cline in the period of rhythm in a beetle”. We have revised the manuscript as per the referee requests and hope the revised version is now suitable for publication in '' PLOS ONE”. We have addressed all of the reviewer’s comments – see our responses below. The author responses to the reviewer’s comments are as follows. Each reviewer’s comment is highlighted in bold, and our responses follow immediately below.

5. Review Comments to the Author

Reviewer #1: This is an interesting ms concerning the geographical variability of the circadian rhythm phenotype in T castaneum. The literature is mixed on this subject with classical Pittendrigh experiments in Drosophila species in Japan suggesting that the amplitude of the rhythm is stronger at higher latitudes because it has to counterbalance the effects of extreme photoperiods which would tend to make animals arrhythmic. There is not much molecular evidence to back this up, indeed recent studies from Helfrich-Foerster suggest the opposite, in that a weaker circadian rhythm in northern latitudes is more flexible for adaptation of circadian behavior to extreme photoperiodic conditions. Costa's group suggests that the circadian light response in D. melanogaster should be reduced at higher latitudes - this pair of papers (Science, 2007???) is not cited. 

Author responses: Thank you for highlighting that our manuscript has potential and for your interesting thoughts on our manuscript. We have followed your suggestions and revised the manuscript following your recommendations. We are appreciate it for your teaching the two important papers which show similar results from our study. We altered the phrase (Lines 69-75) with the references (Lines 376-382).　

Discuss the Helfrich-Foerster and Costa studies on weaker fly circadian rhythms in northern latitudes mentioned by reviewer 1. 

Author responses: Thank you for your teaching the two important papers which show opposite results from our study. We have cited these, and discussed with these studies (Lines 185-211).

Consequently it would have been nice if the study had also included a phase shift response to light as that would have been perhaps more relevant at these latitudes. Nevertheless, this study comes out in favour of the latter hypothesis. The results reflect a lot of work and the ms is clear to read and understand. 

Author responses: Thank you for your useful suggestion. Unfortunately, we are sorry to inform you that it is not possible to conduct this experiment with these wild beetle strains because most of these local populations are not currently being kept in captivity. We're sorry we can't fix it here, but we're very glad you found this study interesting.

My one quibble is that the effects are tiny and require a huge N to be observed. Also the variation in latitude is only 8oC which doesn't help to dissect out an effect. Furthermore the power statistic used is not very sophisticated and there are better methods available, but I suspect the result would be the same. 

Author responses: Thank you for your suggestion. We can't come up with a better way to improve on the status quo, and we think that using other methods would not change the results as you suggested, so we didn't alter this point. Instead, in lines 269-278, we added a discussion of the too large sample size and weak detection effect in this analysis.

The Discussion is a bit too long considering the brevity and conciseness of the results, so perhaps a little less speculation would be in order. Otherwise, the study is OK.

Author responses: We have reduced the amount of consideration according to your and another reviewer's points. Also we removed some phrase to short the discussion of the revised manuscript.

We hope the revised version of the manuscript is suitable for publishing.

Reviewer #2: The authors conducted a study on the effects of geographic (primarily latitudinal) variation on circadian rhythmicity in the red flour beetle. Strengths of the study included the large sample size and rearing the populations for 2 generations in the lab to separate out maternal/environmental effects. While the English was professionally edited, I often found the ideas to be communicated in an imprecise or unclear manner.

Thank you for highlighting that our manuscript has potential and for your interesting thoughts on our manuscript. We have followed your suggestions and revised the manuscript following your recommendations.

1. It was completely unclear why the authors repeatedly referred to longitude early in the manuscript (e.g., L29, L40...a key word!, L47, L53, L78). The study is primarily designed to detect latitudinal clines given the much larger latitudinal range sampled. The authors state that longitudinal clines are also known (L47) but don't give a citation. Based on their titles, all the references cited on L54 (refs 8-12) refer to latitudinal clines. I can understand why they would want to test for this possibility within their data, but I don't know that there is any a priori reason to expect a longitudinal cline. This was also very confusing to me because they referred to altitudinal clines (L26 of abstract) making me think that maybe these would be investigated, but think they were ever discussed in the manuscript. A third point of confusion is that the authors said they were analyzing effects of temperature (L130), but I don't think they ever did.

Author responses: Thank you for your suggestions. “Altitude” in L.26 is my writing miss, and thus I changed it to “longitude”, instead of “altitude”. Sorry for “effects of temperature is also my miss writing, and thus we removed “temperatures” and then altered it to “To analyze the effects if latitude and sex on the period and power of…”. (L 138-139).

2. The discussion on L61-69 is very unclear. The authors seem to be mixing points related to "where flies live" and latitude. By "where flies live" do they mean geographic features that might be distinct from latitude? If so what are they? L67 "pioneering"? Some of the studies cited here are quite old ("pioneering"). Is there a consensus emerging in the newer literature? The discussion overall was hard to follow.

Author responses: We changed the wording that is taken to be geographic distribution to latitude, and we've added a couple of recent papers to the list (Lines 66-75).

3. The predominant finding of the paper was that there is an effect of latitude on amplitude/power of the circadian rhythm. This finding would be more useful if it were put into some kind of context. What is the seasonal variation in photoperiod at the northern vs. southern end of the range? What are the typical temperature ranges? How does this compare with other studies in which circadian clines were or weren't found?

Author responses: Added the context in discussion as you requested (Lines 190-211).

4. While the sample size was apparently quite large, it was described in a very confusing way. For example, L109-110 makes it sound like only one beetle was measured per population. Maybe one was measured per population in each trial and several were monitored at once? L117/TableS1 is confusing because it sounds like it would be the number of beetles measured per population, but I think this is the starting population that was collected and used for breeding (??).

Author responses: Sorry for the confusion caused by our writing style. The sample size of Table S1 shows individual numbers measured for circadian rhythm. Therefore, the number of individuals with measured circadian rhythms is really very, very large. Therefore, we added the phrase “We measured the circadian rhythms of 17 to 128 individual beetles for a single population”, in lines 116-118, and we placed the following phrase “The sample size measured for circadian rhythm of each population is shown in Table S1”, just after the previous sentence.

5. L113 the mention of "a clear Petri dish" again confused me...these are the same petri dishes referred to in L110, right? How many petri dishes were monitored at a time? Was there one animal in each dish? I was really confused. 

Author responses: At a time, we monitored 128 individuals; each animal was put it in each Petri dish. We have prepared 128 locomotor activity sensors. Thus we added the phrase “At a time, we monitored 128 individuals, meaning we are measuring 128 sensors at the same time”, in Lines 123-124.

6. L151-152. This description was inadequate. In what direction was the relationship, and how did it compare with the latitudinal effect on amplitude/power? I think this is important because total activity may affect the power to detect a change in amplitude or period (if animals aren't moving much it's harder to detect a change). Are these trends strong enough that such an indirect effect might be important?

Author responses: Thank you for your helpful comments. We have improved the sentence about the relationship (Lines145-146).

Since there was a weak negative correlation between activity and power of rhythm (Spearman’s correlation coefficient = -0.22), lower activities do not necessarily cause lower rhythm.

Moreover, we conducted path analysis to detect the direct and indirect effects. We found the significant direct effect of latitude on power of rhythm and the similar estimated regression coefficient (-1.84 +- 0.76(SE)) to the one reported in the main text (-2.35 +- 1.08(SE)) although there was the significant indirect effect (latitude -> activity -> power of rhythm).

Therefore, we concluded that higher latitude caused lower rhythm although the effect size was small. Likewise, the result of the period was qualitatively same with the one in the main text.

Altered the phrase to “The results of the regression analysis showed that total activity was associated positively with latitude and sex (Table 1, Figure 4).”

7. L158 this statement is really vague. Are the authors talking about clines or just inter-individual variation? What are they trying to say?

Author responses: Removed it.

8. L166-167 The transition to this sentence is really rough. "On the other hand" would usually indicate a contrast with the previous statement, which isn't the case here.

Author responses: Removed it.

9. L176-177 This sentence seemed like an odd way to end the paragraph. And it is a sort of contrast to the opening sentence (L168-9). Why not combine them into one strong sentence at the beginning. Apparently, a limited number of studies have identified this trend. It's fine to just say that (once) and would be even better if the authors could put it in context. Are there just a few because it's only occasionally studied? Because most studies don't have sufficient power, or because it seems to be rare?

Author responses: We combined the two phrases, as you requested.

10. L188-194 This section again convoluted. L188-190 is a wordy sentence that should be streamlined/clarified. The examples should be clearly organized to indicate which support wide dispersal and which don't. My sense is that both Drury (33) and Arnold (34) suggest limited dispersal. This is obscured by the use of "on the other hand" which implies a contrast. On top of this the text about Semeao's study is repeated twice with very similar text (L195-197 and L233-237).

Author responses: We've put together a short series of these sentences as you requested, and removed the sentence from Semeao's study, which appears a second time, to avoid sentence duplication, as you requested (Lines 240-242).

11. L252 I would argue that not only do they need molecular studies of the circadian rhythmicity (e.g., clines in alleles of specific circadian genes), but also to couple genetic studies (i.e. population assignment) more directly with phenotypic studies.

Author responses: Altered the last phrase as you requested.

Minor

L51 "indicators of circadian rhythms" is vague. These are characteristics of rhythms, and variations in these characteristics could indicate adaptation etc.

Author responses: Altered as you requested (Lines　50-51).

L55 "Additionally" here doesn't make sense. The topic sentence makes it sound like they are switching to talking about "insects other than Drosophila" but the preceding paragraph is also mostly about insects and there is no mention of Drosophila.

Author responses: Altered as you requested (Lines　54).

L58-59 a positive relationship between "latitude and circadian rhythm" doesn't make sense. I think the authors might mean a positive relationship between latitude and period.

Author responses: Altered as you requested (Lines 57-58).

L75 The syntax is hard to follow. I suggest "This species is found throughout most of Japan, except for the northern island of Hokkaido, meaning..."

Author responses: Changed it as you requested (Lines 81-83).

L89-91 Please write the coordinates in a more standard way.

Author responses: Rewrote (Lines 95 and 96).

L101-103 These two sentences almost completely repeat each other.

Author responses: Removed the latter sentence (see Lines 106-107).

L106 "To assess circadian rhythmicity" or "To assess characteristics of circadian rhythms" or "To assess circadian phenotypes"

Author responses: Changed to “"To assess circadian rhythmicity" (Line 110).

L123 periodogram is misspelled.

Author responses: I think “periodogram” is not misspelled. Instead, altered from “periodgram” to “periodogram” in Lines 129 and 131.

L137 power of the rhythm

Author responses: Altered as you requested (Line 139).

L138-139 this description was confusing. It made it sound like they were only considering one (presumably latitude), but in fact they considered each of them in separate analyses.

Author responses: Explained precisely more for this point (Lines 138-139).

L218 "if without human cleaning" doesn't make any sense. I kind of understand that humans cleaning out the pests would disrupt the colonization/selection/evolution, but that's not explained.

Author responses: Thank you for your suggestion, and thus we altered the phrase to “including human cleaning which may affect to colonization, selection and evolution of creatures in each mill” (Lines 240-242).

L236 I might misunderstand, but wouldn't lack of isolation suggest higher gene flow?

Author responses: There was an error in our original manuscript. We altered it as the correct description “the isolation at each mill suggested lower levels of gene flow…” (Lines 258-259).

L242 suggest "is not occurring on the spatial scale that we examined"

Author responses: Changed as you requested (Lines 264-265).

L246 significantly smaller than what? I think the authors just mean significant p values.

Author responses: Altered the phrase to “Notably, the results of the present study showed P-values smaller than 0.05, even for small effect sizes, due to the very large sample size” in Lines 269-270.

L253 I don't know what the authors mean by different phenotypic phenomena than other insect species.

Author responses: Removed it.

Table 1 was misaligned in my version.

Author responses: Rewrote Table 1, and added “Adjusted R-squared”, an index to show the effect size as the editor requested.

Thank you for your useful comments, and we hope the revised manuscript is suitable for publication.

With best wishes, 

Takahisa Miyatake

---

## [Decision Letter · Decision Letter 1]

1 Dec 2020

PONE-D-20-27518R1

Amplitude of circadian rhythms becomes weaker in the north, but there is no cline in the period of rhythm in a beetle

PLOS ONE

Dear Dr. Miyatake,

Thank you for submitting your manuscript to PLOS ONE. After careful consideration, we feel that it has merit but does not fully meet PLOS ONE’s publication criteria as it currently stands. Therefore, we invite you to submit a revised version of the manuscript that addresses the points raised during the review process.

Please fix all minor issues raised by the reviewers.

Further, it is highly recommended to get support from a native speaker or a professional editing service to improve the English of the manuscript. 

We look forward to receiving your revised manuscript.

Kind regards,

Henrik Oster, Ph.D.

Academic Editor

PLOS ONE

Reviewers' comments:

Reviewer's Responses to Questions

**Comments to the Author**

1. If the authors have adequately addressed your comments raised in a previous round of review and you feel that this manuscript is now acceptable for publication, you may indicate that here to bypass the “Comments to the Author” section, enter your conflict of interest statement in the “Confidential to Editor” section, and submit your "Accept" recommendation.

Reviewer #1: All comments have been addressed

Reviewer #2: (No Response)

2. Is the manuscript technically sound, and do the data support the conclusions?

Reviewer #1: Yes

Reviewer #2: Yes

3. Has the statistical analysis been performed appropriately and rigorously? 

Reviewer #1: Yes

Reviewer #2: Yes

4. Have the authors made all data underlying the findings in their manuscript fully available?

Reviewer #1: Yes

Reviewer #2: Yes

5. Is the manuscript presented in an intelligible fashion and written in standard English?

Reviewer #1: No

Reviewer #2: No

6. Review Comments to the Author

Reviewer #1: The ms is interesting but the English really needs to be improved before publication so some editorial assistance is required. Also the organisation of the Discussion could be better. Here are a few further comments.

L66-68. Incorrect statement. The long TIM isoform, not the short one, is negatively correlated with latitude ie higher frequency at lower latitudes.

L203-205 Again the timeless statement is not correct.

‘The frequency of a clock gene ( timeless ) of D. melanogaster studied from 40 to 65 degrees in latitude was negatively phased with latitude, but in populations col lected in southern

Europe by other years , this negative trend seemed to be unclear [21] .’

If the author wishes to use the timeless story, which I think has an important relevant message for this ms. which the authors do touch upon in their discussion of local adaptations and bottlenecks later on, perhaps they might use the paragraph below. I suggest the timeless part be moved to the latter part of the discussion.

‘The frequency of the timeless allele encoding a long TIM isoform of D. melanogaster initially studied from 40 to 65oN was initially negatively correlated with latitude, but in populations collected in extreme southern Europe (32-35oN) this trend was reversed [21] . It turned out that the latitudinal cline was actually caused by natural selection spreading the recently derived tim allele in all directions from a point of origin in southern Italy, thereby generating a distance rather than a latitudinal cline. This ‘distance cline’ was confirmed by studying the tim allele in the Iberian Peninsula (reference Zonato et al 2018 PMID: 29183263) where the cline was reversed. These studies reveal that the history of the genes that encode clinal characteristics also need to be considered when attempting to interpret spatial distributions. Such historical perspectives may account for some of the contradictory results that have been generated in the geographical analyses of biological rhythm phenotypes.

L221 the power of a rhythm is not necessarily the amplitude. It is the general sinusoidal shape of the rhythm. Thus a very sinusoidal but low amplitude rhythms could still have considerable power.

L229 ‘rear flying’????? explain

L291-293. Suggest ‘ Furthermore, given the seasonal differences in light exposure at different latitudes, studying the phase shift of the locomotor rhythm to brief light pulses might also provide a relevant phenotype that might discriminate between northern and southern population, as was the case for Drosophila timeless variants (21)

Reviewer #2: L25 missing word "these traits are" (?)

L55-59 It takes a little "mental gymnastics" for a reader to compare a trend in "higher latitudes" with "a positive relationship with latitude". I suggest adding something to indicate that the two studies are showing trends in the same direction. Like "Similarly, a positive relationship..."

L66 I'm not quite sure what is meant by "frequency of a clock gene". I think the authors mean something like period/frequency of expression. Also not quite sure what is meant by "negatively phased" (negatively correlated?)

L69 I'm not really sure what's meant by "The other context of circadian rhythm". It's not really clear how this is really different from the previous paragraphs where the authors were also talking about relationships with latitude and rhythms. Seasonal variation is mentioned but not really explained.

L71 pioneering

L74 wording doesn't seem right, and it seems trivial to say that the results will be discussed in the context of the literature. Perhaps eliminate this sentence, especially since the next sentence acknowledges the discrepancies.

L82 The authors know more about this than I do, but I usually see this name written as "Hokkaido" in English. Confirm that the spelling is as desired.

L101 this sentence seems to repeat the sentence starting on L91.

L107 Suggest deleting this sentence as it is essentially repeated in the sentence beginning on L109. The sentence beginning on L109 seems better because the association with grain is the part most similar to natural conditions. The temperature and photoperiod presumably vary among sites.

L152-154 I don't understand what the authors are saying here. If you are trying to analyze the relationship between two things (geographical area and power) you need to include them in the same model (?) I think they perhaps mean that effects of latitude and sex were analyzed separately? Or maybe they mean 3 things: latitude, longitude and sex?

L198-199 this is misleading sounds like it spans a range of 32 degrees. Suggest something like "in a range of approximately 8 degrees latitude (from about 32 to 41°N).

L199 Since the authors make it a point that there are no effects of longitude, the longitudinal range (about 10 degrees) should also be given.

L199 The authors say "Previous studies have often" but only cite one study. Preferably cite several examples or refer to the reference in a different way like "(e.g., [21])" (to indicate that it is an example) or ([21] and references therein) if you are citing the reference as a sort of review.

L203 see my comments about similar text within the introduction

L206 what does "this negative trend seemed to be unclear" mean? A negative trend wasn't observed? Or maybe not significant?

L207 it is possible that a cline could be detected

L211 I don't really understand what the authors mean by "survey with closely related species". Do they mean extend the geographic range by pooling results from two species? This idea is unclear.

L211 "In examining the clock cline between different species" is awkward wording. Maybe "In comparing clock clines among speices"

L225-227 wording unclear...kind of repeats "suggest that this beetle disperses very well [35] suggested very high levels of active dispersal" (??) I think the second part of the sentence is meant to be an example. Maybe authors mean "....very well, e.g., Ridley et al. [35] inferred very high levels of active dispersal..."

L229 I don't know what "including rear flying" means

L232; as follows

L248 I'm not sure what the authors mean by ideal environment. I think they mean "ideal" for large effects of bottlenecks and local adaptation. It's not written clearly.

L272 suggest "a cline in circadian amplitude"

L277-279 and L282-286 These two sentences say the same thing. Can they be consolidated?

L288 This line doesn't make sense.

L469 indicate threshold (p-value) for significance.

7. PLOS authors have the option to publish the peer review history of their article (what does this mean?). If published, this will include your full peer review and any attached files.

Reviewer #1: No

Reviewer #2: No

---

## [Author Response · Author response to Decision Letter 1]

18 Dec 2020

The author responses to the reviewer’s comments are as follows. Each reviewer’s comment is highlighted in bold, and our responses follow immediately below.

Please fix all minor issues raised by the reviewers.

We fixed all issued raised by the reviewers.

Further, it is highly recommended to get support from a native speaker or a professional editing service to improve the English of the manuscript. 

The English of the re-revised manuscript has been checked by a professional native English editing service.

6. Review Comments to the Author

Reviewer #1: The ms is interesting but the English really needs to be improved before publication so some editorial assistance is required. Also the organisation of the Discussion could be better. Here are a few further comments.

We are glad that you are interested in this paper. The English of the re-revised manuscript has been checked by a professional native English editing service.

L66-68. Incorrect statement. The long TIM isoform, not the short one, is negatively correlated with latitude ie higher frequency at lower latitudes.

Sorry for our mistake. We corrected it as you requested (Lines 66-69).

L203-205 Again the timeless statement is not correct.

‘The frequency of a clock gene ( timeless ) of D. melanogaster studied from 40 to 65 degrees in latitude was negatively phased with latitude, but in populations col lected in southern

Europe by other years , this negative trend seemed to be unclear [21] .’

If the author wishes to use the timeless story, which I think has an important relevant message for this ms. which the authors do touch upon in their discussion of local adaptations and bottlenecks later on, perhaps they might use the paragraph below. I suggest the timeless part be moved to the latter part of the discussion.

‘The frequency of the timeless allele encoding a long TIM isoform of D. melanogaster initially studied from 40 to 65oN was initially negatively correlated with latitude, but in populations collected in extreme southern Europe (32-35oN) this trend was reversed [21]. It turned out that the latitudinal cline was actually caused by natural selection spreading the recently derived tim allele in all directions from a point of origin in southern Italy, thereby generating a distance rather than a latitudinal cline. This ‘distance cline’ was confirmed by studying the tim allele in the Iberian Peninsula (reference Zonato et al 2018 PMID: 29183263) where the cline was reversed. These studies reveal that the history of the genes that encode clinal characteristics also need to be considered when attempting to interpret spatial distributions. Such historical perspectives may account for some of the contradictory results that have been generated in the geographical analyses of biological rhythm phenotypes.

Sorry for our mistake, again. We rephrased and moved it to the later part of the discussion as you requested (Lines 291-303) and added the reference (Lines 468-471). 

L221 the power of a rhythm is not necessarily the amplitude. It is the general sinusoidal shape of the rhythm. Thus a very sinusoidal but low amplitude rhythms could still have considerable power.

Removed “power” (Line 217)

L229 ‘rear flying’????? explain

Reword it as “short-range flying” (Line 225)

L291-293. Suggest ‘ Furthermore, given the seasonal differences in light exposure at different latitudes, studying the phase shift of the locomotor rhythm to brief light pulses might also provide a relevant phenotype that might discriminate between northern and southern population, as was the case for Drosophila timeless variants (21)

Thank you so much for your suggestion for the last phrase. We rephrased it as you requested (Lines 285-289).

Reviewer #2: L25 missing word "these traits are" (?)

Added “are” (Line 25).

L55-59 It takes a little "mental gymnastics" for a reader to compare a trend in "higher latitudes" with "a positive relationship with latitude". I suggest adding something to indicate that the two studies are showing trends in the same direction. Like "Similarly, a positive relationship..."

Added “Similarly,“ (Line 57).

L66 I'm not quite sure what is meant by "frequency of a clock gene". I think the authors mean something like period/frequency of expression. Also not quite sure what is meant by "negatively phased" (negatively correlated?)

Altered to “The frequency of the timeless allele encoding a long TIM isoform,” (Lines 66-67), and to “negatively correlated” (Line 68-69), as you requested. Thank you for your suggestions.

L69 I'm not really sure what's meant by "The other context of circadian rhythm". It's not really clear how this is really different from the previous paragraphs where the authors were also talking about relationships with latitude and rhythms. Seasonal variation is mentioned but not really explained.

We removed the phrase (Line 69).

L71 pioneering

Re-worded (Line 70).

L74 wording doesn't seem right, and it seems trivial to say that the results will be discussed in the context of the literature. Perhaps eliminate this sentence, especially since the next sentence acknowledges the discrepancies.

Eliminated the sentence (Lines 73-74).

L82 The authors know more about this than I do, but I usually see this name written as "Hokkaido" in English. Confirm that the spelling is as desired.

You are quite right. (Line 81).

L101 this sentence seems to repeat the sentence starting on L91.

Removed the sentence (see Line 100).

L107 Suggest deleting this sentence as it is essentially repeated in the sentence beginning on L109. The sentence beginning on L109 seems better because the association with grain is the part most similar to natural conditions. The temperature and photoperiod presumably vary among sites.

Removed the sentence (see Line 106).

L152-154 I don't understand what the authors are saying here. If you are trying to analyze the relationship between two things (geographical area and power) you need to include them in the same model (?) I think they perhaps mean that effects of latitude and sex were analyzed separately? Or maybe they mean 3 things: latitude, longitude and sex?

Sorry to confuse you. The two things were latitude and longitude. Therefore, we rewrote this phrase as “To avoid multicollinearity between latitude and longitude (the correlation coefficient (r) between them was 0.83), we didn’t include both simultaneously in the model as an explanatory variable but constructed a separate model with each value.” (Lines 149-152).

L198-199 this is misleading sounds like it spans a range of 32 degrees. Suggest something like "in a range of approximately 8 degrees latitude (from about 32 to 41°N).

Altered it as you suggested (Lines 196-197). Thank you.

L199 Since the authors make it a point that there are no effects of longitude, the longitudinal range (about 10 degrees) should also be given.

We gave it in Line 201.

L199 The authors say "Previous studies have often" but only cite one study. Preferably cite several examples or refer to the reference in a different way like "(e.g., [21])" (to indicate that it is an example) or ([21] and references therein) if you are citing the reference as a sort of review.

Rephrased it as “([21] and references therein)” (Line 197-198).

L203 see my comments about similar text within the introduction

Rewrote and moved it to the last of the discussion as you and the reviewer 1 commented (Lines 291-303).

L206 what does "this negative trend seemed to be unclear" mean? A negative trend wasn't observed? Or maybe not significant?

We have corrected this sentence as noted by reviewer 1. (Lines 197-201).

L207 it is possible that a cline could be detected

Added “a” (Line 203).

L211 I don't really understand what the authors mean by "survey with closely related species". Do they mean extend the geographic range by pooling results from two species? This idea is unclear.

We rewrote it as “it may be possible to survey the point if we use a closely related species, such as Tribolium confusum who inhabits also in Hokkaido.” Is this make sense? If not the case, we will remove the phrase at the next revision. (Lines 206-208).

L211 "In examining the clock cline between different species" is awkward wording. Maybe "In comparing clock clines among speices"

Changed it as you requested. (Line 208).

L225-227 wording unclear...kind of repeats "suggest that this beetle disperses very well [35] suggested very high levels of active dispersal" (??) I think the second part of the sentence is meant to be an example. Maybe authors mean "....very well, e.g., Ridley et al. [35] inferred very high levels of active dispersal..."

Altered as you requested. (Line 221-222).

L229 I don't know what "including rear flying" means

Altered to “short-range flying” (Line 225).

L232; as follows

Changed (Line 228).

L248 I'm not sure what the authors mean by ideal environment. I think they mean "ideal" for large effects of bottlenecks and local adaptation. It's not written clearly.

Reworded (Lines 244-245).

L272 suggest "a cline in circadian amplitude"

Added “circadian” as you suggested (Line 267).

L277-279 and L282-286 These two sentences say the same thing. Can they be consolidated?

Thank you for your advice. The two sentences have slightly different meanings and are difficult to concatenate. I hope you understand it (See 274-281). 

L288 This line doesn't make sense.

Reworded it (Line 283).

L469 indicate threshold (p-value) for significance.

Added p-values in the captions of Figures 2 (p<0.05) and 4 (p<0.01), for lines 482-491, respectively. 

Thank you for all of your kind suggestions and advices. Red text, other than where the reviewers have noted, is where a professional native English editing service has made corrections. We hope the revised version of the manuscript is now suitable for publication.

With best wishes, 

Takahisa Miyatake

Behalf of all authors

Masato S. Abe, Kentarou Matsumura, Taishi Yoshii

---

## [Decision Letter · Decision Letter 2]

23 Dec 2020

Amplitude of circadian rhythms becomes weaken in the north, but there is no cline in the period of rhythm in a beetle

PONE-D-20-27518R2

Dear Dr. Miyatake,

We’re pleased to inform you that your manuscript has been judged scientifically suitable for publication and will be formally accepted for publication once it meets all outstanding technical requirements.

Kind regards,

Henrik Oster, Ph.D.

Academic Editor

PLOS ONE

Additional Editor Comments (optional):

n/a

Reviewers' comments:

Reviewer's Responses to Questions

**Comments to the Author**

1. If the authors have adequately addressed your comments raised in a previous round of review and you feel that this manuscript is now acceptable for publication, you may indicate that here to bypass the “Comments to the Author” section, enter your conflict of interest statement in the “Confidential to Editor” section, and submit your "Accept" recommendation.

Reviewer #1: All comments have been addressed

2. Is the manuscript technically sound, and do the data support the conclusions?

Reviewer #1: Yes

3. Has the statistical analysis been performed appropriately and rigorously? 

Reviewer #1: Yes

4. Have the authors made all data underlying the findings in their manuscript fully available?

Reviewer #1: Yes

5. Is the manuscript presented in an intelligible fashion and written in standard English?

Reviewer #1: Yes

6. Review Comments to the Author

Reviewer #1: The ms is much improved although the English still needs some work. L217-218 is redundant so cut. the effects on power are small but significant and fit with other findings on other species. Given the Results section is so short and concise, could the authors not similarly reduce the length of the Discussion which is a little repetitive and overlong and a little rambling.

7. PLOS authors have the option to publish the peer review history of their article (what does this mean?). If published, this will include your full peer review and any attached files.

Reviewer #1: No

---

## [Editor Report · Acceptance letter]

2 Jan 2021

PONE-D-20-27518R2 

Amplitude of circadian rhythms becomes weaken in the north, but there is no cline in the period of rhythm in a beetle 

Dear Dr. Miyatake:

I'm pleased to inform you that your manuscript has been deemed suitable for publication in PLOS ONE. Congratulations! Your manuscript is now with our production department. 

Kind regards, 

on behalf of

Prof. Henrik Oster 

Academic Editor

PLOS ONE